# Spectrum of Bacterial Pathogens from Urinary Infections Associated with Struvite and Metabolic Stones

**DOI:** 10.3390/diagnostics13010080

**Published:** 2022-12-28

**Authors:** Adam Halinski, Kamran Hassan Bhatti, Luca Boeri, Jonathan Cloutier, Kaloyan Davidoff, Ayman Elqady, Goran Fryad, Mohamed Gadelmoula, Hongyi Hui, Kremena Petkova, Elenko Popov, Bapir Rawa, Iliya Saltirov, Francisco Rodolfo Spivacow, Belthangady Monu Zeeshan Hameed, Katarzyna Arkusz, Alberto Trinchieri, Noor Buchholz

**Affiliations:** 1Private Medical Center “Klinika Wisniowa”, Department of Urology, 65-417 Zielona Góra, Poland; 2Department of Clinical Genetics and Pathomorfology, University of Zielona Góra, 65-417 Zielona Góra, Poland; 3Urology Department, Alkor HMC, Hamad Medical Corporation, Al Khor P.O. Box 3050, Qatar; 4Department of Urology, IRCCS Ca’ Granda Ospedale Maggiore Policlinico, University of Milan, 20122 Milan, Italy; 5Department of Urology, Faculty of Medicine, CHU de Québec, Laval University, Québec, QC G1V 0A6, Canada; 6Acibadem City Clinic Tokuda Hospital, 1000 Sofia, Bulgaria; 7Urology Department, Assiut University, Assiut 71515, Egypt; 8Shar Teaching Hospital, Sulaymanyah 46001, Kurdistan Region, Iraq; 9Department of Urology, Renji Hospital, Shanghai Jiaotong University School of Medicine, Shanghai 200025, China; 10Department of Urology and Nephrology, Military Medical Academy, 1000 Sofia, Bulgaria; 11Smart Health Tower, Sulaymaniyah 46100, Kurdistan Region, Iraq; 12Instituto de Investigaciones Metabólicas, IDIM Department of Urology, Buenos Aires C1012AAA, Argentina; 13Department of Urology, Kasturba Medical College, Manipal 576104, Karnataka, India; 14Department of Biomedical Engineering, Faculty of Mechanical Engineering, University of Zielona Góra, 65-417 Zielona Góra, Poland; 15U-Merge Ltd. (Urology for Emerging Countries), 14561 Athens, Greece

**Keywords:** urinary calculi, urinary tract infection, struvite, urease, *Escherichia coli*, *Proteus mirabilis*, Gram-positive pathogens

## Abstract

Objective: The purposes of this multi-center study were to evaluate the rate of infection stones and to evaluate the urine cultures of patients with infection stones. Materials: Charts of adulpatients with urinary stones were reviewed and data on stone analyses and urine cultures were collected. Results: In total, 1204 renal stone formers (RSFs) from 10 countries were included (776 males, 428 females). Fifty-six patients (4.6%) had struvite stones. The highest frequency of struvite stones was observed in India (23%) and Pakistan (18%). Lower rates were reported in Canada (2%), China (3%), Argentina (3%), Iraq (3%), Italy (3.5%) and Poland (3%), and intermediate rates in Egypt (5.5%) and Bulgaria (5.4%). Urine cultures were retrieved from 508 patients. Patients with struvite stones had a positive culture in 64.3% of the samples and patients with other stones, in 26.7%. In struvite stones, the most common isolates were *Escherichia coli* (27.7%) and *Proteus* spp. (27.7%), followed by *Klebsiella* spp. (16.7%); in other types of stone, it was *Escherichia coli* (47.6%), followed by Gram-positive bacteria (14.0%) Conclusions: The struvite stone composition was associated with a urinary infection, although an infection was not demonstrable with a conventional midstream urine culture in about 30%.

## 1. Introduction

Urinary tract infections can change the composition of urine, causing an increased risk of the formation of several types of urinary stones. Bacterial strains with urease activity, otherwise known as urea-splitting bacteria, can break down urea, causing an increase in the urinary concentrations of ammonium and bicarbonate in the presence of water. The increase in urinary bicarbonate causes the alkalinization of the urine and, together with the increase in urinary ammonium, increases the urinary saturation for struvite or magnesium ammonium phosphate (MgNH_4_PO_4_·6H_2_O) and carbonate apatite [Ca_10_(PO_4_)_6_CO_3_] [1].

In addition, highly alkaline urine and a high urinary ammonia concentration act locally by damaging the glycosaminoglycans that make up the superficial layer of the urothelium to protect cells from a bacterial invasion [2,3]. A bacterial biofilm is thus formed on the surface of the urothelium of the pelvis and renal calyxes, in which struvite and apatite crystals precipitate [4,5]. The bacteria, in turn, produce extracellular polysaccharides and lipopolysaccharides, contributing to the rapid growth of the stone by the subsequent rapid apposition of the layers of the mineral material mixed and held together by the organic material [6,7,8,9,10]. A dust of crystalline material is formed around the bacteria, which tends to precipitate around the bacteria and inside the bacteria themselves, with the formation of microliths after bacteriolysis [11,12]. These mechanisms explain the fast growth of these types of stones, which tend to fill all the kidney cavities to form a cast of the cavities and take on a coralliform or staghorn shape. Struvite is observed almost exclusively in association with a urinary tract infection caused by urease-producing bacteria [13,14,15]. Carbonate apatite is a mineral phase that can be observed even in the absence of an infection. A few authors, with an accurate analysis of the crystalline composition by infrared spectroscopy, have shown that it is possible to distinguish carbonate apatite originating from a urine infection from a non-infectious one. The evaluation of a higher degree of carbonation, defined as the ratio between the intensity of the band of carbonate ions at 1420 cm^−1^ and the intensity of the band of phosphate ions at 1035 cm^−1^, allows the recognition of those stones that have an infectious origin. The carbonation rate was significantly greater (22%) in patients with a visible bacterial imprint using electron microscopy than in those without (8%) [16]. However, the overall predictive value of highly carbonated apatite composition stones for culture-positive stones was not great [17].

Ammonium urate stones are related to high urinary levels of ammonium and uric acid that may occur with both metabolic conditions and infections by urea-splitting bacteria [18,19,20]. High urinary levels of ammonium and uric acid are observed in patients with a history of inflammatory bowel disease, ileostomies, laxative use or abuse or in morbidly obese patients. In patients with recurrent urinary infections, ammonium urate stone formation can occur when excessive levels of ammonium combine with enough acid urates in the urine.

The purposes of this study were to evaluate the rate of infection stones in a large series of renal stone formers observed in different countries of the world and to evaluate the urine cultures of patients with infection stones.

## 2. Materials and Methods

In 2020, U-merge, an association gathering urologists from all over the world, launched a study to collect the results of urinary stone analyses among different populations in the countries of its members. The charts of adult patients (>18 years) with renal or ureteral stones observed in each participating center were reviewed. The data of the patients who had undergone a chemical analysis of the stones available were collected. Any method of stone analysis was accepted, but the methodology had to be known and registered. The age, gender, stone composition and nationality of each patient were recorded in an Excel database. A minimum number of 30 patients per center was required.

After the publication of the data in July 2021 [21], the Scientific Office of U-merge decided to extend the study to the association of infections with stones of different compositions. Patients were subdivided according to the stone composition in patients with infection stones and patients with metabolic stones. The stones—analyzed by wet chemical, infrared spectroscopy and X-ray diffractometry—were classified as infection stones when they were composed of struvite (>50%). Stones of carbonate apatite (>50%) or ammonium urate (>50%) were separately considered. The charts of the patients were re-checked to retrieve the results of the urine cultures. The rates of positive urine cultures in patients with infections and metabolic stones were computed. The spectrum of bacterial isolates was also registered.

The Statistical Package for the Social Sciences (SPSS) version 11.5 for Windows was used for the statistical analysis. Comparisons were considered to differ significantly if *p* < 0.05.

## 3. Results

In total, 1204 renal stone formers (RSFs) were included (776 males, 428 females) from 12 institutions of 10 countries (Argentina, Bulgaria, Canada, China, Egypt, India, Iraq, Italy, Pakistan and Poland). Of those, 56 (4.6%) patients had struvite stones, 15 (1.2%) had carbapatite stones and 7 (0.58%) had ammonium urate stones. The mean age of the patients with struvite stones (45.6 ± 15.0 years) was significantly lower (*p* = 0.049) than in the patients with metabolic stones (49.5 ± 14.4 years). The male-to-female ratio was not significantly different in the patients with infection stones compared with the patients with metabolic stones (*n* = 33/23 vs. 743/405; *p* = 0.37) (Figure 1).

The highest frequency of struvite stones was observed in India (23%) and Pakistan (18%) whereas the lowest were reported in Canada (2%), China (3%), Argentina (3%), Iraq (3%), Italy (3.5%) and Poland (3%). Intermediate rates were found in Egypt (5.5%) and Bulgaria (5.4%) (Figure 2). Carbapatite stones were reported from Canada (20%), Iraq (5%) and Bulgaria (2%). Ammonium urate stones were observed in a few cases (N = 6; 3.5%) in Bulgaria.

The mean age of the patients with struvite stones from European countries (51.6 ± 16.7 years) was higher than the mean age of the patients from the areas including Egypt, Iraq, Pakistan and India (41.2 ± 13.6 years) as well as other countries (41.8 ± 10.2 years) (*p* = 0.042). The M/F ratio was clearly in favor of males in the areas including Egypt, Iraq, Pakistan and India whereas it was slightly in favor of females in European countries and in other countries (*p* = 0.032) (Figure 3A,B).

Urine cultures were retrieved from 508 patients. The cultures were positive in 18/28 patients (64.3%) with struvite stones and in 130/482 (26.7%) patients with other types of stones. The spectrum of isolates is shown in Table 1 and Figure 4. The spectrum of bacterial pathogens was significantly different in the patients with struvite stones compared with the patients with other stones. For the 18 microbial isolates in the patients with struvite stones, the most common ones were *Escherichia coli* (27.7%) and *Proteus* spp. (27.7%), followed by *Klebsiella* spp. (16.7%), Gram-positive bacteria (5.5%) and *Pseudomonas aeruginosa* (5.5%). These bacteria accounted for 83.3% of the total flora tested. 

In the other types of stones, *Escherichia coli* was the most frequent isolate (47.6%), followed by Gram-positive bacteria (14.0%) and *Klebsiella* spp. (7.8%). *Proteus* spp. and *Pseudomonas aeruginosa* accounted for only 4.6% and 2.3%, respectively. The uropathogens in the Gram-positive group were *Enterococcus faecalis* (*n* = 13), *Enterococcus faecium* (*n* = 1), *Staphylococcus aureus* (*n* = 2), *Staphylococcus cepra* (*n* = 1), *Streptococcus agalactiae* (*n* = 1) and other rare Gram-positive bacteria (*n* = 1). Other isolates included other Gram-negative bacteria, *Candida* spp., *Ureaplasma urealyticum* and mixed flora.

## 4. Discussion

Forty years ago, infection stones were the second most frequent type of kidney stones, making up about 15–20% of all cases of urinary stones [22,23]. Over the years, the frequency of infected stones has progressively decreased due to improvements in health conditions that have made the diagnosis and treatment of urinary infections in women more effective, the introduction of non-invasive stone treatments that spare the integrity of the urinary tract and the increase in calcium-oxalate calculi. In several countries, a decrease in the frequency of struvite stones has been shown by studies performed over different time periods in the same country. In Japan, the frequency of struvite stones decreased from 1965 to 2005 in both men (from 7.5% to 1.4%) and women (from 23.3% to 5.1%) [24]; in Spain, it progressively decreased from 12.5% in 1979 to 9.8% in 1987 and to 6.7% in 1998 [25]; in Italy, it decreased from 24% in the period 1981–95 [26] to a rate ranging between 1.6 and 6.5% in the period 2003–2018 [27]; in Germany, the frequency decreased from 4.9% to 3.3% in males and from 13.5% to 9.2% in females between 1977 and 2006 [28]; in Australia, it decreased from 14% in the 1970s to 12% in the 1980s and to 7% in the period 2009–2011 [29]; and finally, in the United States, the frequency of struvite stones decreased from 7.8 to 3.0% between 1990 and 2010 in females whereas, in the same interval, it remained stable from 2.8 to 3.7 in males [30]. Other studies have confirmed a frequency of struvite calculi of less than 5% in most countries of the world [31,32,33,34,35].

Our data confirmed that today, in most countries, infection stones account for less than 5% of stone cases, although higher percentages are still observed in several countries where there are pockets of poverty-related substandard hygiene in the population. 

We observed the highest frequency of struvite stones in India (23%) and Pakistan (18%). A high frequency of struvite stones is associated with a high prevalence of urinary tract infections in these geographical areas. Studies have shown that in large areas of India there are hygienic and sanitary conditions that favor a high prevalence of urinary tract infections, especially in women. In poorer settings of India, menstrual hygiene management practices can be particularly unhygienic for girls and women, depending on the socioeconomic status, education level, local traditions and beliefs and access to water and sanitation resources [36]. Furthermore, there are no facilities for the screening and diagnosis of urinary tract infections at peripheral government health centers. The rate of urinary tract infections among pregnant women attending antenatal clinics of district hospitals is high [37]. Finally, studies have highlighted the increasing rate of antimicrobial resistance among uropathogens in the community settings of India [38]. Similarly, a high frequency of urinary tract infections has been observed in Pakistan among pregnant women and women attending gynecology clinics [39,40].

Traditionally, infection stones are more frequent in the female population whereas metabolic stones are more frequent in the male population. A few recent studies also showed a higher frequency of struvite stones in women [41,42,43]. Our data confirmed that the male-to-female ratio tended to be higher in patients with metabolic stones than in those with struvite stones. Surprisingly, the number of male patients with infection stones was, however, slightly higher than the number of female patients with struvite stones. This finding, if confirmed by other larger series, could be suggestive of a change in the risk factors for infection stones. On the other hand, in a series from Germany it was shown that the female-to-male ratio changed from 1:0.61 to 1:0.95 between 1977 and 2006, respectively, in patients with struvite stones [28]; in Australia, the contemporaneous peak age for struvite stones in 2009–2011 was a group of 61–70-year-old men whereas in the 1980s it was the female 21–30 age group [29]. The most common causative pathogen for urinary tract infections (UTIs) is *Escherichia coli* [44], although the spectrum of pathogens observed in patients with UTIs associated with urinary stones has been characterized by lower rates of *Escherichia coli*, at around 40%, and higher rates of *Proteus* spp. and *Klebsiella* spp. In a study of almost a thousand microbial isolates from patients with a UTI associated with upper tract urinary stones, the most common isolate was *Escherichia coli* (41.3%), followed by Gram-positive bacteria (25.1%), the KES group (*Klebsiella* spp., *Enterobacter* spp. and *Serratia* spp.) (14.2%), *Proteus* spp. (11.7%) and *Pseudomonas aeruginosa* (4.1%) [45]. This different distribution of pathogens is due to the impact of infection stones that are associated with urea-splitting bacteria infections, which are predominately *Proteus* spp. and, to a lesser extent, *Klebsiella* spp. and staphylococci; *Escherichia coli* rarely presents with urease activity [23,41,46]. Nevo et al. [47] compared patients who had stones with a struvite content increasing from 1–25% to 76–100%. In patients with a lower struvite content, the preoperative urine culture was positive in 31% and *Escherichia coli* was the most frequent isolate. In stones with a higher struvite content, the urine culture was positive in 90% and the most frequent isolate was *Proteus* spp. (47%). These observations were confirmed by studies that evaluated the results of intraoperative cultures of the stones, showing high rates of urease-producing bacteria in association with struvite stones [48,49,50].

In accordance with previous findings, our results confirmed an *Escherichia coli* infection rate of approximately 40% in patients with urinary stones in general, but a lower rate of *Escherichia coli* of 27.7% in the group of patients with struvite stones. A finding of our study that deserves to be exhaustively discussed is the finding of a negative urine culture in a considerable proportion of patients with struvite stones. This finding agreed with the observations of other authors, who failed to demonstrate the presence of bacteria in the urine and stones of several patients with struvite calculi [41,49,50,51].

The role of an infection by urease-producing bacteria in the pathogenesis of struvite stones is indisputable; however, in most series it has not been possible to demonstrate the presence of classically described urease-splitting bacteria in all the urinary samples of patients with struvite stones. This finding can be explained in several ways. A midstream urine culture might not be representative of the microbiology of the urine from the renal pelvis or the stone itself. Mariappan et al. [52] demonstrated that a positive stone culture and a pelvic urine culture were better predictors of potential urosepsis than bladder urine and observed that, in a few cases, a pelvic culture and/or a stone culture was positive in patients with a negative midstream culture. A recent meta-analysis [53] confirmed that a stone culture and a renal pelvic urine culture are more reliable than a midstream urine culture in identifying causative organisms and directing the antibiotic therapy of UTIs that developed after percutaneous renal surgery. A negative culture can also be explained by the presence of urease-producing bacteria that do not grow in common urine cultures such as Ureaplasma urealyticum and Corynebacterium urealyticum [54,55]. Finally, a negative stone culture can be the result of a previous effective antibiotic treatment. Parkhomenko et al. [50] found that two-thirds of struvite stone patients with negative stone cultures presented a history of at least one previous positive culture for urea-splitting bacteria. The colonization of struvite stones with organisms thought to be non-urease-producing could be explained by a secondary colonization/infection of the stone with non-urease-producing bacteria and the plasmid-mediated acquisition of the genes responsible for urease production.

Our study had a few limitations. The number of patients with struvite stones and urine cultures was lower in comparison with other series from Northern America, Central Europe and China [23,40,41,42,45,46,48,49], although it was higher than in series from other, less developed, countries [51,56]. Moreover, we did not report about the size of the stones, which can be useful to better define the characteristics of struvite stones. The analysis of the stones was not performed with infrared spectroscopy in all the centers, so the presence of a minor component of struvite may have escaped in a few series, with a consequent underestimation of the frequency of struvite stones. The microbiological investigation was limited to the culture of midstream urine without performing an intraoperative culture of the stone, which would have allowed the diagnosis of the presence of an infection in a greater number of cases. These limitations were inherent in the very design of the study, which aimed to include centers from all over the world and represented their usual diagnostic standards. The data from our study confirmed the reduction in the frequency of struvite stones in most countries and a tendency toward the disappearance of a greater frequency of struvite stones in women. However, the results obtained must be confirmed by prospective studies with a homogeneous protocol and a more extensive microbiological investigation, including the culture of the stone.

## 5. Conclusions

Infection stones are becoming less frequent in all countries of the world, although they still represent a challenging clinical condition. The struvite stone composition was associated with a urinary infection, although in about 30% of cases, an infection was not demonstrable with a conventional midstream urine culture. In patients with clinical and radiological signs suggestive of a struvite stone, even if the urine culture was negative, the risk of a postoperative infection and sepsis should be always considered because of the frequent association of an infection with multi-drug-resistant bacteria [41] and enterococci [49], requiring a broad-spectrum coverage.

## Figures and Tables

**Figure 1 diagnostics-13-00080-f001:**
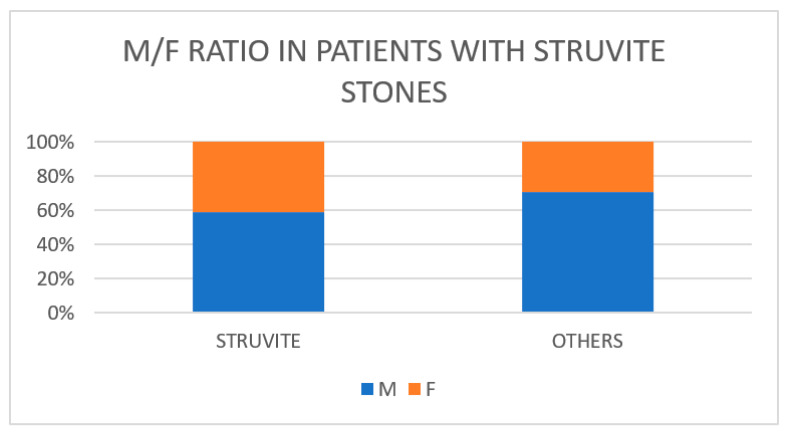
Male-to-female ratio of patients with struvite stones.

**Figure 2 diagnostics-13-00080-f002:**
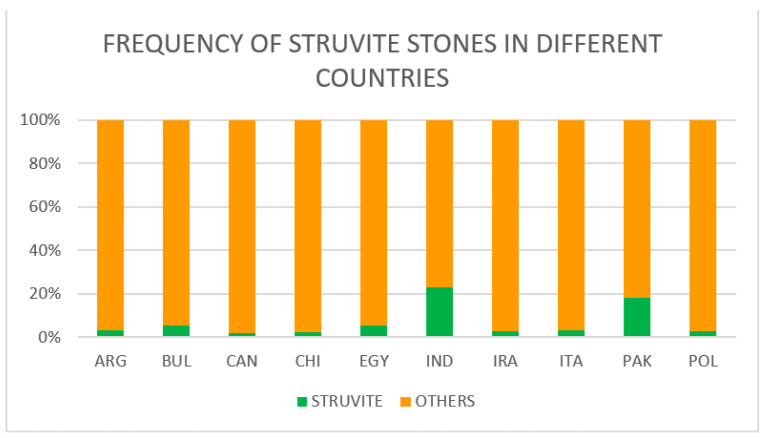
Rates of struvite stones in stone formers from different countries.

**Figure 3 diagnostics-13-00080-f003:**
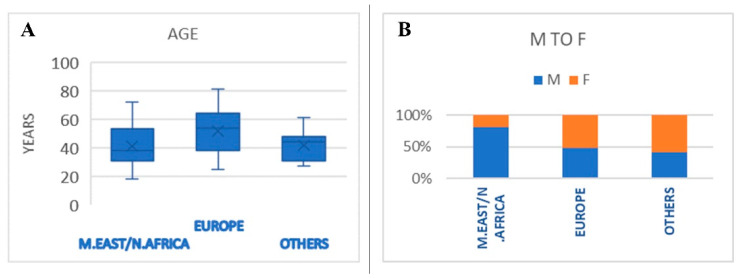
Mean age (**A**) and M/F ratio (**B**) of patients with struvite stones from different geographical areas.

**Figure 4 diagnostics-13-00080-f004:**
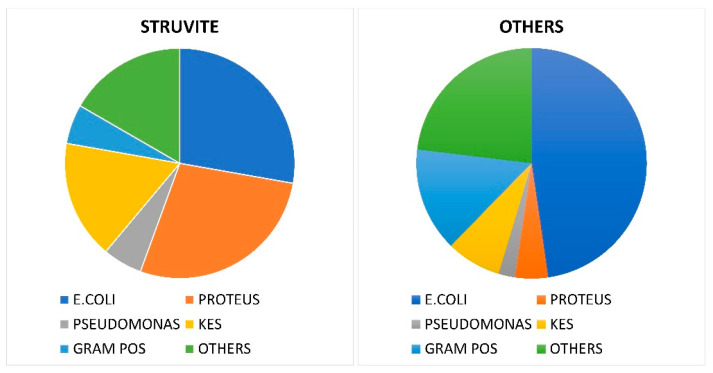
Bacterial spectrum of urine of patients with struvite stones (>50%) and other stones of metabolic origin.

**Table 1 diagnostics-13-00080-t001:** Spectrum of isolates associated with stones of different compositions.

	*E. coli*	*Proteus*	*Pseud*	*Klebs*	Gram-Pos	Others	Neg	Tot
Struvite	5	5	1	3	1	3	10	28
	27.7%	27.7%%	5.5%	16.7%	5.5%	16.7%	35.7%	
Others	62	6	3	10	19	30	352	482
	47.6%	4.6%	2.3%	7.7%	14.6%	23.0%	73.1%	

*Pseud*: *Pseudomonas a*. *Klebs*: *Klebsiella* spp.

## Data Availability

The datasets analyzed during the current study are available from U-merge upon reasonable request.

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
