# Peer review of "Spectrum of Bacterial Pathogens from Urinary Infections Associated with Struvite and Metabolic Stones"

_diagnostics, 2022, doi:10.3390/diagnostics13010080_

Round 1
Reviewer 1 Report (Previous Reviewer 2)
I have the impression that the discussion is too long.
Author Response
Welcome
In attachment you can find our response to your comments
Best regards

Reviewer 2 Report (Previous Reviewer 1)
Within the resubmission the authors answered the concerns raised by previous reviews among which clinical significance of the study has been stressed.
The study is well-designed. The discussion reads well. References are relevant.
Author Response
Welcome
We thank the Reviewer for the time spent to review our paper and for kind comments.
Best regards
Reviewer 3 Report (New Reviewer)
Dear authors,
Thank you for submitting this article to Diagnostics. The authors have done an excellent job putting together the statistics on the rate of infection stones.
However, this would have been better if this was revised to a review paper instead of an article. The article has information presented from across the globe but does not present any novelty.
With many statistics involved, can the authors provide statistical significance for the data presented?
The article has serious formatting issues (ex: lines 61,64, 72, etc).
Author Response
Welcome
In attachment you can find our response to your comments
Best regards

This manuscript is a resubmission of an earlier submission. The following is a list of the peer review reports and author responses from that submission.
Round 1
Reviewer 1 Report
The authors present a multi-centre epidemiology study on the rate of infection stones. The study is well-designed and the results are interesting, esp with he number of struvite stones decreasing in most countries due to improved health conditions. Statistical analysis is kept simple and clear. The discussion reads well. The authors are also aware of the limitations of their study. Judging by the dates of publication of the references, it seems that collecting up-to date data was necessary. The only thing I might lack is stressing the correlation of the acquired data with every-day clinical practice.
Author Response
We thank the Reviewers for the time spent to review our paper and for their useful comments and
suggestions we tried to comply in the following point by point reply.
Changes are highlighted in red in the text.
collecting up-to date data was necessary
Recent literature was searched on PubMed by using the string “struvite AND (microbiology OR infection)” for the last 5 years. A few references were found and added.
Some recent studies also showed a higher frequency of struvite stones in women (26-28).
26 Gao X, Lu C, Xie F, Li L, Liu M, Fang Z, Wang Z, Ming S, Dong H, Shen R, Sun Y, Peng Y, Gao X. Risk factors for sepsis in patients with struvite stones following percutaneous nephrolithotomy. World J Urol. 2020; 38:219-229.
27 Zhang D, Li S, Zhang Z, Li N, Yuan X, Jia Z, Yang J. Urinary stone composition analysis and clinical characterization of 1520 patients in central China. Sci Rep. 2021; 11:6467.
28 Popovtzer B, Khusid JA, Bamberger JN, Lundon D, Gallante B, Sadiq AS, Atallah W, Lifshitz D, Gupta M. Do Infection-Associated Stone Subtypes Behave the Same Clinically? A Retrospective Bi-center Study. J Endourol. 2021 Dec 16. doi: 10.1089/end.2021.0460. Epub ahead of print.
and Gao et al (26) found an equal frequency of 18.64% for both E.coli and Proteus spp. Interestingly, Nevo et al (32) compared patients who had stones with struvite content increasing from 1-25% to 76-100%. In patients with a lower struvite content, pre-operative urine culture was positive in 31% and E. coli was the most frequent isolate. In stones with higher struvite content, urine culture was positive in 90% and the most frequent isolate was Proteus spp (47%).
32 Nevo A, Shahait M, Shah A, Jackman S, Averch T. Defining a clinically significant struvite stone: a non-randomized retrospective study. Int Urol Nephrol. 2019; 51:585-591.
every-day clinical practice
We added this sentence to the Conclusions to highlight suggestions for clinical practice.
Infection stones are becoming less frequent in all countries of the world, although they still represent a challenging clinical condition. Struvite stone composition is associated to urinary infection although in about 30% of cases infection is not demonstrable with conventional midstream urine culture. In patients with clinical and radiological signs suggesting a struvite stone, even if urine culture is negative, the risk of postoperative infection and sepsis should be always considered because of the frequent association with infection by multi-drug resistant bacteria (26) and enterococci (35) requiring broad-spectrum coverage.
Reviewer 2 Report
I think it is better to put "struvite" in the title of the paper, and in the text "Struvite stone is also expressed as MAP (magnesium ammonium phosphate) stone".
Figure 3 shows the age, scrutiny graph, and bacterial spectrum graph. Please Correct.
In Figure 3 for the age, a boxplot is better than a bar graph.
There is no explanation for the abbreviation for bacteria in the figure showing bacterial spectrum.
In the Conclusion, "Struvite stone composition is an indicator of urinary infection", did you mention that the proportion of struvite stones varies from country to country?
Reviewer 3 Report
1.The author's language expression needs further polishing .
2.For this report, the sample size should be larger to better draw an accurate conclusion.
3.Data on stones may require more detailed analysis, such as stone size. The existing conclusions of this paper lack innovation.
Author Response
We thank the Reviewers for the time spent to review our paper and for their useful comments and suggestions we tried to comply in the following point by point reply.
Changes are highlighted in red in the text.
The language was revised by our authors living in English-speaking countries. We considered 56 patients with struvite stones out from a larger population of 1204 renal stone patients, although we were able to obtain microbiology in only 28 of them. We agree that the number of our series is not high compared to other series from Northern America, Europe or China which included from 34 to 125 struvite stone patients with urine or stone cultures ((Bichler et al n=128, Popovtzer et al n=79, Zhang et al n=115, Paonessa et al n=125, Parkhomenko et al n=30, Gao et al n=97, Flannigan et al,=121 Nevo et al n=34). However, the number of the very few series reporting data from other countries are even lower (Tavichakorntrakool et al n=8, Shahait et al n=6). We also agree that stone size can be an adjunctive information that unfortunately we have not collected. Consequently, we added in the limitations of the study the following sentences.
The number of patients with struvite stones and urine culture was lower in comparison to other series from Northern America, Central Europe, and China (13, 26-28,31,32,34,35) although higher than in series from other less developed countries (36,41). Moreover, we did not report about the size of stones than can be useful to better define the characteristics of struvite stones. 41.Tavichakorntrakool R, Prasongwattana V, Sungkeeree S, Saisud P, Sribenjalux P, Pimratana C, Bovornpadungkitti S, Sriboonlue P, Thongboonkerd V. Extensive characterizations of bacteria isolated from catheterized urine and stone matrices in patients with nephrolithiasis. Nephrol Dial Transplant. 2012;27:4125-30.
As we stated in the limitations “These limitations are inherent in the very design of the study which aimed to include centers from all over the world representing their usual diagnostic standards.”
On the other hand, some original observations were reported in the paper as follows
“today,in most countries, infection stones account for less than 5% of stone cases, although higher percentages are still observed in some countries where there are pockets of poverty-related sub-standard hygiene in the population”
“the number of male patients with infection stones was however slightly higher than the number of female patients with struvite stones. This finding, if confirmed by other larger series, could be suggestive of a change of the risk factors for infection stones”
Consequently we added a sentence in the limitations.
In fact, data from our study confirmed the reduction in the frequency of struvite stones in most countries and a tendency to the disappearance of the greater frequency of struvite stones in women.
Reviewer 4 Report
This interesting paper describes the spectrum of bacterial pathogens from urinary infections associated with infection and metabolic stones.
The study has some strong positive aspects. However, it also has a significant limitation that leads me to reject it for publication.
On the positive side, the subject of the study is both interesting and useful, in that kidney stones are a common problem and investigating associated infections has the potential to increase our knowledge of their pathogenesis. The sample is more than sufficiently large. The multicentric nature of the study, drawn from all around the globe, makes the results less dependant on a particular racial group or culture.
You identified the main limit of the study: your analysis was made only on patients’ urine and did not investigate the stones themselves. However, an analysis of the stones is fundamental to identifying the range of infections that can arise and giving results for urine alone severely impairs the usefulness of your otherwise commendable study.
Author Response
We thank the Reviewers for the time spent to review our paper and for their useful comments and suggestions we tried to comply in the following point by point reply.
Changes are highlighted in red in the text.
As we have already pointed out in the discussion, the lack of data on the microbiological examination of the stones is a limitation of this study. Unfortunately, stone analysis is not a routine diagnostic procedure in patients undergoing procedures for the removal of urinary stones. The retrospective design of our study therefore did not allow for the collection of data relating to kidney stone culture. It is desirable that this information can be evaluated in the future with a prospective study involving the culture of the stone in all patients studied, as we stated in the Discussion.
Round 2
Reviewer 2 Report
The manuscript has been properly revised.